# Antibody Mediated Intercommunication of Germinal Centers

**DOI:** 10.3390/cells11223680

**Published:** 2022-11-19

**Authors:** Theinmozhi Arulraj, Sebastian C. Binder, Michael Meyer-Hermann

**Affiliations:** 1Department of Systems Immunology, Braunschweig Integrated Centre of Systems Biology, Helmholtz Centre for Infection Research, 38106 Braunschweig, Germany; 2Institute for Biochemistry, Biotechnology and Bioinformatics, Technische Universität Braunschweig, 38106 Braunschweig, Germany

**Keywords:** germinal centers, antibody feedback, germinal center intercommunication, germinal center shutdown, broadly neutralizing antibodies, mathematical modeling

## Abstract

Antibody diversification and selection of B cells occur in dynamic structures called germinal centers (GCs). Passively administered soluble antibodies regulate the GC response by masking the antigen displayed on follicular dendritic cells (FDCs). This suggests that GCs might intercommunicate via naturally produced soluble antibodies, but the role of such GC–GC interactions is unknown. In this study, we performed in silico simulations of interacting GCs and predicted that intense interactions by soluble antibodies limit the magnitude and lifetime of GC responses. With asynchronous GC onset, we observed a higher inhibition of late formed GCs compared to early ones. We also predicted that GC–GC interactions can lead to a bias in the epitope recognition even in the presence of equally dominant epitopes due to differences in founder cell composition or initiation timing of GCs. We show that there exists an optimal range for GC–GC interaction strength that facilitates the affinity maturation towards an incoming antigenic variant during an ongoing GC reaction. These findings suggest that GC–GC interactions might be a contributing factor to the unexplained variability seen among individual GCs and a critical factor in the modulation of GC response to antigenic variants during viral infections.

## 1. Introduction

Antibody responses result in the generation of memory B cells and long-lived plasma cells from structures called germinal centers (GCs) in secondary lymphoid organs [1]. GCs are composed of two compartments called dark and light zones (DZ and LZ, respectively) [2,3]. In the DZ, B cells divide and mutate their antibody genes by a process termed somatic hypermutation [4]. In the LZ, B cells are then selected based on the ability to capture immune complexes, receiving survival signals from follicular dendritic cells (FDCs) [5] and T follicular helper (Tfh) cells [6,7], that polarize towards B cells with highest peptide density [8]. While the lack of selection leads to B cell apoptosis [9,10], selected cells acquire different fates such as differentiation into precursors of memory/plasma cells [11,12,13] or recycling to the DZ for cell divisions [14,15,16,17]. Repeated rounds of cycling between the GC zones when accompanied by mutation and selection leads to affinity maturation of B cells, thus progressively increasing the affinity of antibodies produced [18]. 

Several mechanisms act to regulate the magnitude and efficiency of GC responses. Antigen amount has been suggested to impact antibody production [19], affinity maturation [20] and affect the trade-off between quality and quantity of GC responses [21,22]. Modulating the antigen administration dynamics can enhance the GC response and promote the development of broadly neutralizing antibodies [23,24,25]. Dynamics of antigen presentation due to cycling in FDCs [26,27] has been predicted to determine the trade-off between stability of immune complexes and accessibility to GC B cells [28]. In addition, passively administered and endogenously produced soluble antibodies have been shown to impact the GC kinetics and affinity maturation by masking the antigen presented on FDCs [29]. This phenomenon of modulation of antigen accessibility in GCs by soluble antibodies is termed as antibody feedback. Previous studies have suggested that high antibody feedback accelerates the shutdown [29] and impairs the efficiency of GC reaction [21]. Furthermore, it has been suggested that GCs might intercommunicate and regulate each other by the exchange of endogenously produced soluble antibodies [29]. 

Regulation of GCs by soluble antibodies is also extensively studied in the context of developing broadly neutralizing antibodies. Administering antibodies targeting immunodominant epitopes can suppress the GC response to these epitopes and shift the focus of GCs towards rare epitopes, thus opening up possibilities for therapeutic interventions [30]. However, the role of GC–GC interactions by endogenously produced soluble antibodies is underexplored due to technical challenges in specifically targeting GC–GC interactions and whether such interactions promote a natural shift in epitope recognition is unknown. Our previous study [21] suggested that strong antibody feedback can terminate GC responses earlier and reduce plasma cell production. There was a tendency for antibody feedback to enhance the affinity of B cells at early time points, but the affinity maturation process was terminated at an early stage due to quick shutdown of GCs which reduced the ultimately achieved affinity of plasma cells produced. Similarly, soluble antibodies from early formed GCs also impaired a late GC suggesting that the extent of GC inhibition might vary depending on the strength of feedback and timing of GC onset [21]. As the reverse feedback of soluble antibodies from late GCs to early GCs was ignored in the previous study, the impact of antibody feedback on the collective GC response in a lymphoid organ could not be inferred. In this study, we performed in silico simulations of a more realistic scenario where multiple asynchronous GCs interact by the exchange of soluble antibodies under conditions of different antigen complexities and delivery kinetics. 

## 2. Materials and Methods

A simulation framework was developed (schematically shown in Figure 1A) that allows to investigate the interaction of multiple co-existing GC reactions in silico under the influence of antibodies produced by plasma cells from those GC reactions. This framework was built on a previously developed agent-based GC model [31,32,33] and its extension to simulate multiple asynchronous GCs [34].

### 2.1. Overview of GC Model

Each GC was modelled as a 3D lattice with CXCL12/CXCL13 chemokine distributions [33] and different cell types including FDCs, Tfh cells and B cells. GC area was equally divided into DZ and LZ. FDCs (200 in number) were randomly placed in the LZ region. Each FDC was assumed to occupy multiple connected lattice sites, where the soma of each FDC gives rise to six dendrites of length 40 µm each. A fixed initial antigen amount of 3000 portions per FDC was considered and the antigen was uniformly distributed to the different lattice nodes occupied by the same FDC. FDCs were assumed to be static. Tfh cells (200 in number) were randomly distributed in the GC area, that eventually tended to accumulate in the LZ region due to chemotaxis towards CXCL13.

Founder B cells were incorporated into the lattice at a rate of 2 cells per hour for a duration of 96 h. Shape space [35] was used for BCR specificity representation where the position of a B cell with respect to a predefined optimal position (antigen) was considered as a measure of binding probability. The dimension of the shape space was chosen as 4, since it is suitable for recapitulating the extent of affinity maturation and a decrease in the population of GC B cells at late timepoints without additional assumptions [15]. Ten positions (from 0–9) were considered along each dimension of the shape space. For founder B cells with randomly chosen affinities, any of the shape space positions were randomly assigned by uniform sampling resulting in average low affinity. For high-affinity founder cells, shape space positions were randomly sampled at a distance of 2 mutations from the optimal position. Nine mutations away from the optimal position could be considered as a rough estimate of the lower bound for affinity, which is consistent with experimentally observed number of mutations [36,37]. While the shape space positions are uniformly sampled to randomly select BCR affinities, antigens are fixed at positions away from boundaries of shape space to avoid potential artifacts. Mutation of BCRs beyond the boundaries is prohibited. Each founder cell divides six times before acquiring LZ phenotype. Mutations were modeled as random shift in the shape space position of B cells to a neighboring site and a mutation probability of 0.5 per division was assumed [4,38].

B cells acquiring LZ phenotype were initially assigned a state *Unselected*, which corresponds to cells yet to be selected by FDCs and Tfh cells. During a fixed time period called antigen collection phase of 0.7 h, *Unselected* cells interacting with FDCs capture antigen with a probability dependent on the BCR affinity. While failure to collect antigen within the antigen collection phase leads to the *Apoptotic* state, successful antigen capture leads to the state *FDCselected*. *FDCselected* cells search for Tfh cells for a duration of 3 h. Each B cell—Tfh cell interaction lasts for 6 min. As multiple B cells are allowed to interact with a Tfh at the same time, Tfh cells polarize towards and deliver signals to the B cell with highest antigen uptake. After the interaction period, B cells are allowed to interact with other Tfh cells, and the signals acquired from each interaction is integrated. B cells that have acquired Tfh signals for a total duration of 30 min progress to the *Selected* state. Failure to meet this requirement leads to the *Apoptotic* state.

*Selected* cells acquire the DZ phenotype and divide, where the number of divisions were assumed to be dependent on the antigen uptake. Seventy-two percent of these divisions were asymmetric ([39] and Appendix A in [31]) in terms of antigen distribution and the daughter cell retaining the antigen differentiates into an *Output* cell. Alternatively, the other daughter cell and the cells that underwent symmetric divisions switch to the *Unselected* state and acquire a LZ phenotype. *Output* cells differentiate into plasma cells with the rate calculated from a half-life of 24 h.

### 2.2. Simulation of Multiple GCs

Multiple GCs were simulated where each GC follows the basic characteristics described above. To reduce the computational complexity, GCs initialized at similar time points were assumed to behave in an identical manner and were approximated by a single representative GC as described in [34]. Eight representative GCs were simulated where the number of GCs represented by each simulated GC and GC onset times were estimated by fitting the initial phase of GC number kinetics data [40] with a hill function (Appendix A in [34]). Alternatively, in simulations with synchronous GC onset, 8 representative GCs were simulated each representing 10 GCs and all the simulated GCs were initialized at the start of the simulation.

### 2.3. Antibody Feedback and GC–GC Interactions

Plasma cells from the simulated GCs secrete antibodies with a rate constant *k*_p_. Antibodies with different affinities were classified into 11 bins (i) that differ in the dissociation rate constant ($k_{\mathrm{off}}\left(i\right)$) such that the dissociation constant $K_{d}\left(i\right)$ varies between 10^−5.5^ and 10^−9.5^ M. Antibodies were assumed to undergo degradation with a rate constant of $k_{\deg}$ = ln (2)/14 days^−1^. Antibodies are assumed to homogeneously spread throughout the organism over a volume V of 10 mL which accounts for the volume of fluid in circulatory and lymphatic systems [41] with a slightly higher margin to account for the clearance mechanisms and antigen internalization by cells not explicitly considered in this study.

Antibody concentration $A_{r}\left(i\right)$ in each bin i from the $r^{\mathrm{th}}$ GC is calculated as
(1)$$\frac{dA_{r}\left(i\right)}{dt}=\frac{k_{p}}{V}n_{p}\left(i\right)-k_{\deg}A_{r}\left(i\right)$$
where $k_{p}$ is the antibody production rate and $n_{p}\left(i\right)$ is the number of plasma cells with affinity corresponding to bin i.

The fraction of FDC antigen masked by soluble antibodies $C_{\mathrm{FDC}}$ was determined by the chemical kinetics equation
(2)$$\frac{dC_{\mathrm{FDC}}\left(i\right)}{dt}=k_{\mathrm{on}}G_{\mathrm{FDC}}\overset{8}{\underset{r=1}{\sum }}A_{r}\left(i\right)N_{r}-k_{\mathrm{off}}\left(i\right)C_{\mathrm{FDC}}\left(i\right)$$

The antibody concentration was multiplied with $N_{r}\;$ which corresponds to the number of GCs represented by the  r^th^ simulated GC, to account for the contribution from GCs that were not explicitly simulated. While the binding/unbinding of antibodies were assumed to decrease/increase the concentration of FDC antigen $G_{\mathrm{FDC}}$ accessible to B cells, changes in concentration of antibodies was neglected. The strength of GC–GC interactions was varied by modulating the antibody production rates in Equation (1). Low, medium and strong interaction strength correspond to antibody production rates $k_{p}=$ 3 × 10^−18^ [42], 10^−17^ and 2 × 10^−17^ mol per hour per cell, respectively. A control simulation with no interaction between GCs (no feedback) was also considered, where the soluble antibodies were not allowed to mask the antigen on FDCs. In control simulations, both GC–GC interactions via soluble antibodies and the feedback on individual GCs due to self-produced antibodies were ignored.

The immune power (IP) [21,43] was calculated by estimating the fraction of test antigen of concentration R = 10^−5^ M bound to the soluble antibodies. IP was normalized with respect to $k_{p}$ corresponding to low interaction strength ($k_{\mathrm{ref}\;}=3\times 10^{-18}\;\mathrm{mol}\;\mathrm{per}\;\mathrm{hour}\;\mathrm{per}\;\mathrm{cell}$) to account for differences in antibody production rates:(3)$$R_{\mathrm{bound}}\left(i\right)=\frac{R\overset{8}{\underset{r=1}{\sum }}A_{r}\left(i\right)N_{r}}{K_{d}\left(i\right)+R}$$
(4)$$IP=\frac{k_{\mathrm{ref}\;}\sum R_{\mathrm{bound}}\left(i\right)}{k_{p\;}R}$$

Simulations were performed using C++ [44] and results were visualized using R [45] and ggplot2 [46]. Each set of simulation was repeated 30 times. C++ code and R scripts used in this study are available at https://doi.org/10.5281/zenodo.6568368 (uploaded on 21 May 2022).

## 3. Results

### 3.1. GC–GC Interactions Decrease the Longevity of GC Responses

Firstly, we investigated whether GC–GC interactions inhibited only individual GC reactions or also the overall GC response. We simulated GCs that were initialized asynchronously (Figure 1B, see Section 2.2) with a single epitope (shape space position 3333, see methods) and founder cell positions randomly chosen anywhere in the shape space giving rise to a wide range of B cell affinities. Longevity of collective GC responses decreased earlier with increasing strength of GC–GC interactions (Figure 1C). In simulations without GC–GC interactions, a small decrease in the antigen resource on FDCs due to consumption by GC B cells impacts the average number of GC B cell divisions and promotes GC shutdown [47]. GC–GC interactions accelerate GC shutdown by reducing the antigen accessibility of B cells. Consequently, total number of plasma cells produced from the GC response was decreased (Figure 1D). Average affinity of plasma cells was also decreased due to the earlier termination of the affinity maturation process (Figure 1E), suggesting that the overall GC response was reduced. The inhibitory influence of soluble antibodies on the overall GC response was also found when the simulated GCs were initialized synchronously (Appendix A, see Section 2.2). This implies that the qualitative impact of GC–GC interactions on the overall GC response does not rely on the extent of synchronicity in GC initiation. A similar decrease in the longevity of GCs, quality and quantity of output were also seen in simulations with multiple epitopes (Appendix A). These results suggested that longevity and output of the overall GC response were limited by antibody-mediated GC–GC interactions.

### 3.2. Late Formed GCs Are Highly Sensitive to Antibody Feedback

In the network of GCs with asynchronous onset, individual GCs showed differences in sensitivity to antibody mediated inhibition and the sensitivity increased with the delay of GC initiation (Figure 1B,F,G). When compared to the early formed GCs, late formed GCs had a stronger reduction in the lifetime, maximum size, number and affinity of plasma cells with more intense antibody-feedback (Figure 1B,F,G). In contrast, individual GCs when initialized synchronously had a similar reduction in the above-mentioned readouts (Appendix A). These results predict that the late GCs are more sensitive to the inhibition by soluble antibodies due to antibody-mediated GC–GC interactions.

In the absence of GC–GC interactions, variability observed in the size of individual GCs at any given time point is primarily due to asynchronous GC onset (Figure 1B), but the lifetime of GCs, overall quality and quantity of plasma cells throughout the GC lifetime were similar (Figure 1B,F,G). Differences in the timing of GC initiation and interaction between GCs led to differences in the kinetics and lifetime of individual GCs. However, the time of shutdown of the different GCs was synchronized by antibody-mediated GC–GC interactions (Figure 1B).

### 3.3. High Affinity Founder Cells Can Counteract Antibody Mediated Inhibition

As the exact mechanisms contributing to asynchronous onset and founder cell composition of GCs is unknown, we tested whether the sensitivity of late GCs to antibody mediated inhibition was reduced if the late GCs were seeded by high affinity founder cells. Such high affinity founder cells could be memory cells produced from early GCs. We chose high affinity founder cells (distance of 2 mutations from the optimal clone 3333 in the shape space) for GCs 7 and 8 that were initialized at late time points. The founder cell affinity of other GCs was randomly chosen resulting in relatively lower average affinity as in the previous simulations. Higher affinity founder cells partly counteracted the reduction in maximum size (Figure 2A), plasma cell production and affinity of plasma cells from late GCs (Figure 2E,F). As GCs with high affinity founders were inherently short-lived, longevity of the overall GC response was not enhanced (Figure 2B). Increased plasma cell production and affinity from late GCs had a very small impact on the overall GC response (Figure 2C,D), since the GCs represented by late simulated GCs were relatively few in number compared to the early GCs (see methods).

This result was derived for a single epitope in a primary GC response. Although it is not known whether memory B cells produced from early GC reactions might participate in late GCs of a primary immune response, our results suggest that late GCs can overcome a strong antibody feedback when founded by high-affinity memory cells when compared to lower affinity naïve B cells.

### 3.4. GC–GC Interactions Induce Diversity in Epitope Recognition among Individual GCs

Given the potential role of GC–GC interactions in generating variability in the lifetime, kinetics and affinity maturation of individual GCs, we investigated whether GC–GC interactions also lead to differences in affinity maturation to individual epitopes in the presence of multiple epitopes. We tested the affinity of plasma cells at the end of the GC response from individual GCs in the presence of two epitopes (shape space positions 3333 and 5555) in equal proportion with founder cell positions randomly chosen anywhere in the shape space. With no interaction between GCs, plasma cells from individual GCs had similar affinities to both the epitopes, suggesting that individual GCs support affinity maturation to diverse epitopes. GC–GC interactions increased differences in plasma cell affinity between the two epitopes in individual GCs (Figure 3). Late formed GCs tend to have a higher affinity towards one of the epitopes, while the affinity towards the other epitope remained very low. As both epitopes are equally accessible, GCs did not always attain higher affinity towards the same epitope. (Appendix A). Therefore, such biased affinity maturation towards one of the epitopes was not observed in plasma cell affinity to individual epitopes when averaged with multiple simulation results (Appendix A). These findings are consistent with a higher mutation distance between the two epitopes in shape space (Appendix A). The effects of GC–GC interactions were qualitatively unaffected by changes in the antibody half-life, plasma cell differentiation time, recycling probability and antigen concentration (Appendix A). However, these parameters influenced the quantitative effects of GC–GC interactions. Thus, antibody feedback disturbs the equal chances of one epitope to dominate in individual GC reactions but such differences in epitope recognition on the scale of the overall GC response might not be observable.

### 3.5. GC–GC Interactions Induce Better Affinity Maturation of Rare Epitope in Late Formed GCs

Injecting soluble antibodies against the immunodominant epitope has been shown to shift the focus of in silico GC reactions towards the sub-dominant or rare epitope [30]. To investigate whether a similar shift in the GC response towards rare epitope occurs due to GC–GC interactions, we varied the proportion of two epitopes, with shape space positions 3333 (90%) and 5555 (10%) such that the latter mimics a rare epitope. Founder cell positions were randomly chosen anywhere in the shape space. Interestingly, even though the late GCs had a reduced response towards the dominant epitope, the average affinity of plasma cells towards the rare epitope was slightly higher compared to the early GCs (Figure 4), when the GC–GC interaction strength was increased.

We tested whether the small increase in affinity maturation towards rare epitope enhanced the GC efficiency (IP), which is a combined measure of quality and quantity of plasma cells produced (see methods). The IP of the overall GC response towards the rare epitope was not enhanced by GC–GC interactions (not shown). This result was robust against variation of the total amount of available antigen. Due to the higher sensitivity of late GCs to soluble antibodies, these GCs are short-lived, and their production of plasma cells is greatly diminished. Thus, a small increase in affinity of plasma cells was insufficient to enhance the efficiency of late GCs towards neutralizing the rare epitope.

### 3.6. The GC Founder Cell Composition Determines the Impact of Antibody-Mediated GC–GC Interactions

Previous sections showed that GC–GC interactions induce differences in kinetics and epitope specificity of individual GCs. We have previously predicted that GCs induced in a primary immune response upon immunization with sheep red blood cells have highly heterogeneous lifetimes that could be explained by differences in founder cell composition and antigen availability among individual GCs [34]. To test the impact of antibody-mediated interactions on GCs with different founder cell compositions, we varied the founder cell composition of GCs in silico with two epitopes (shape space positions 2222 and 6666) in equal proportion. Founder cells of GCs 1–3 were chosen randomly giving rise to diverse specificities and founder cells of the rest of the GCs were assumed to be specific to epitope 2 (distance of 1 mutation from shape space position 6666). In the absence of GC–GC interaction, GCs with random founders focused equally on both the epitopes and the average plasma cell affinities of these GCs were similar towards both epitopes (Figure 5E). However, in the presence of GC–GC interaction, GCs 1–3 had a biased affinity maturation towards epitope 1 (Figure 5E). This is due to the high concentration of soluble antibodies against epitope 2 from GCs 4–8, that induced a shift in the focus of GCs 1–3 towards epitope 1 (Figure 5E). These results show that depending on the founder cell composition of interacting GCs, antibody-mediated GC–GC interaction can focus GCs on a particular epitope despite similar accessibility and presence of founder cells recognizing multiple epitopes. Although, the longevity of the overall GC response and plasma cell production were decreased by GC–GC interactions (Figure 5A,B), the total affinity of plasma cells was slightly enhanced towards epitope 1 (Figure 5C) and decreased towards epitope 2 (Figure 5D). Affinity of plasma cells from GCs 1–3 towards both epitopes showed higher variability compared to GCs 4–8 as reflected in the standard deviation (Figure 5E) because in GCs 1–3 founder cells with varying specificities and lower affinities randomly increased their affinities towards one of the two epitopes whereas founder cells of GCs 4–8 had higher affinity only towards epitope 2. Collectively, the total plasma cell affinity (from all GCs) towards epitope 1 showed slightly higher variability compared to epitope 2 (Figure 5C,D).

Taken together, our results suggest that depending on the antigen complexity, antibody-mediated GC–GC interactions lead to differences in individual GC kinetics, quality and quantity of plasma cells in an initiation time dependent manner or a characteristic shift in affinity maturation towards specific epitopes.

### 3.7. Persistent Addition of Antigen Can Overcome the Effects of GC–GC Interactions

Antibody-mediated decrease in the longevity of GC responses and output production might act as a self-regulatory mechanism that is limiting the magnitude of GC responses. As this mechanism acts by limiting antigen accessibility on FDCs, we tested whether a persistent deposition of antigen on FDCs is able to overcome the effects of GC–GC interactions on the overall GC response. Starting from the simulation with low GC–GC interaction and synchronous GC onset (as in Appendix A), we simulated the addition of antigen at a constant rate during the GC response and varied the rate of antigen addition in silico. A single epitope was considered (shape space position 3333) with founder cell positions randomly chosen anywhere in the shape space. Increasing the rate of antigen deposition on FDCs increased the longevity of GCs (Figure 6A). Moreover, persistent addition of antigen increased the maximum GC size, number and affinity of plasma cells (Figure 6B–F). Therefore, persistent antigen addition was able to counteract the inhibition of the GC response by antibody-mediated GC–GC interactions.

### 3.8. GC–GC Interactions Modulate Responses to Antigenic Variants

We hypothesized that GC–GC interactions modulate the GC responses to an incoming antigenic variant during viral infections. We initialized GCs synchronously with a single epitope at shape space position 3333 and persistently added an antigenic variant (epitope 2) from day 7 of the GC reaction. Relatedness of epitope 2 to epitope 1 was varied by choosing shape space positions at a distance of 2, 4 or 8 mutations (at shape space positions 3344, 4444 and 5555) from epitope 1.

Irrespective of the GC–GC interaction strength and relatedness of epitope 2 to epitope 1, average affinity of plasma cells towards epitope 1 was not enhanced compared to simulations with no GC–GC interactions at day 25 of the GC reaction (Appendix A and Appendix A). This suggests that GC–GC interactions tend to suppress affinity maturation towards the existing epitope. When the variant added is very closely related to existing epitope (2 mutations away), no enhancement was observed in affinity maturation towards epitope 2 due to GC–GC interactions (Figures S6 and 7B). With a mutation distance of 4, an increase in GC–GC interaction strength until a moderate interaction strength, resulted in an enhanced plasma cell affinity towards epitope 2 at day 25 of the GC reaction (Figure 7D). However, further increase in the GC–GC interaction strength did not support affinity maturation towards epitope 2 but was rather suppressive. An increase in mutation distance to 8, representing an increased variability between epitope 1 and 2 led only to a weak increase in plasma cell affinity towards epitope 2 (Figure 7F). A very low concentration of soluble antibodies is insufficient to effectively block the existing epitope while a very high concentration disrupts the affinity maturation towards both epitopes by promoting very quick shutdown of GCs (Appendix A). Thus, there is an optimal range for GC–GC interaction strength depending on epitope relatedness that would facilitate affinity maturation towards the antigenic variant.

Increase in the affinity of plasma cells towards epitope 2 was only transient, and over a long-term remained lower than the plasma cell affinity in simulations without GC–GC interaction due to accelerated shutdown of GCs (Appendix A). These findings suggest that shutdown of GCs is a critical factor that hinders an efficient affinity maturation towards antigenic variants.

## 4. Discussion

Modulation of GCs by soluble antibodies is an important strategy for developing broadly neutralizing antibodies [29,30]. However, the natural influence of soluble antibodies by GC–GC interactions is unexplored and would be of importance in better understanding the regulation of GC kinetics and affinity maturation under conditions of different antigen complexity. To investigate the impact of GC–GC interactions under different conditions, we performed in silico simulations of interacting GCs. We analyzed the impact of GC initiation dynamics on GC–GC interactions by considering two extreme cases - asynchronous and synchronous onset of GCs. In asynchronous onset, new GCs were initialized until 12 days after immunization which is based on experimentally observed GC number kinetics [40]. We predicted that GC–GC interactions acted as a self-regulatory mechanism limiting the longevity and magnitude of GC responses, irrespective of the GC initiation dynamics, number of epitopes or founder cell composition considered.

At the level of individual GCs, GC–GC interaction led to variability in lifetime, kinetics and affinity maturation of GC reactions due to differences in GC onset times. GC initiation dynamics impacted the variability between single GCs by modulating GC–GC interactions. Given that the GC initiation dynamics might depend on experimental settings [34], we expect that the extent of variability between single GCs due to intercommunication will also vary depending on experimental conditions. Current estimates of antibody production rates vary greatly [48]. In addition, variability in the GC dynamics and lifespan observed under different experimental conditions [49] suggest a large variability in the dynamics of plasma cell production. Consequently, we expect that the strength of GC–GC interactions would differ depending on immunization conditions. Therefore, the results presented in this study only suggest qualitative effects of GC–GC interactions. Accurate quantification of antibody concentrations and plasma cell dynamics during the course of the GC reaction under different experimental settings would enable a more quantitative and antigen-specific prediction of the effects of GC–GC interactions.

Behavior of late GCs was highly sensitive to the output of early GCs due to communication by soluble antibodies. It is important to note that the sensitivity of late GCs might vary depending on the plasma cell production dynamics of early GCs, which contributes to the early pool of soluble antibodies influencing other GCs. A temporal shift in the memory and plasma cell production during the GC response has been described [13], but the contribution of individual GCs to the memory and plasma cell production is unknown. We predicted that late GCs support better affinity maturation towards rare epitope due to GC–GC interactions by soluble antibodies. Emergence of higher affinity plasma and memory cells to the rare epitope by antibody feedback might promote a higher immune efficiency in a secondary response.

In silico late GCs attained a larger size in the presence of memory B cells as founders when compared to naïve B cells. It is unknown whether the memory B cells from early GCs participate in late GCs formed within the response to a primary immunization and needs future investigation. In the case of secondary GC responses with multiple epitopes, memory B cells recognizing the immunodominant epitope experience higher inhibition by antibody feedback compared to naïve B cells specific to a different epitope [30]. IgM+ memory B cells preferentially participated in secondary GC reactions as opposed to IgG+ memory B cells that had a greater tendency to differentiate into plasma cells upon re-stimulation [50]. Mesin et al. showed that only a limited fraction of memory B cells are able to participate in GC responses to a secondary antigen challenge [51]. This observation might be explained by the antibody-mediated GC–GC interaction that specifically competes with the memory BCR.

In addition to the exchange of soluble antibodies and potential exchange of memory B cells, GCs might also communicate with each other by the exchange of Tfh cells [52]. The role of such Tfh exchange is presently unknown and needs to be investigated in this context in the future.

Previously, we have suggested that GC lifetimes and output could vary among individual GCs within the same lymphoid organ and that these differences cannot be solely explained by GC–GC interactions [34]. Our present study suggests that GC–GC interactions might at least act as a contributing factor to heterogeneous GC lifetimes and affinity maturation in addition to factors such as initial antigen amount and founder cell composition. We speculate that the variability between individual GCs might be required to get a diverse pool of memory cells in preparation for the next challenge with a possibly mutated antigen.

GC–GC interactions might be a major factor regulating antigen availability and limiting GC lifetime. Antibody feedback has been shown to limit booster responses to vaccination against malaria at sub-protective titers but promote the diversification of B cell responses to subdominant epitopes [53]. Our simulations show similar effects with GC–GC interactions by endogenous soluble antibodies suggesting that GC–GC interactions might also pose a challenge in enhancing overall antibody titers to vaccination despite the tendency to promote the development of broadly neutralizing antibodies. As persistent addition of antigen counteracted the inhibition by antibody feedback, minimal antibody feedback might be a contributing factor to the observed longevity of viral infection-induced GCs due to the persistent presence of viral antigen. Similarly, the enhanced magnitude of GC responses seen in the case of slow delivery immunization [23,25] might also be due to reversal of the effects of GC–GC interactions in addition to other factors. It can be speculated that antibody feedback also plays a role in directing the GC response to specific antigenic variants in GC reactions induced by mutating pathogens, thus, avoiding the GC response being trapped in the sense of the original antigenic sin theory. We find that GC–GC interactions can support or hinder the affinity maturation towards the variant depending on the feedback strength and relatedness to existing epitope. Our results also suggests that there exists an optimal GC–GC interaction strength that facilitates the affinity maturation towards an incoming antigenic variant. As only a transient increase in affinity of plasma cells to the antigenic variant was observed due to accelerated GC shutdown, we propose that shutdown of GCs is an important target to enhance the affinity maturation towards antigenic variants. In this context, future studies might explore the GC responses to persistent deposition of different antigenic variants to identify factors that promote the development of broadly neutralizing antibodies.

Testing the effects of GC–GC interactions identified in this study by experimental modulation of the antibody secretion or stability would help verify the predictions. For example, persistent addition of antigenic variant with varying degree of similarity to the existing antigen would enable detection of changes in B cell affinity maturation due to GC–GC interactions. As the effects predicted vary depending on conditions such as whether there is a persistent deposition of antigen, extent of synchronicity between GC initiation times and antigen complexity, experiments will have to be performed under different immunization conditions. The characteristic differences in the behavior of individual GCs predicted in this study might also facilitate the experimental analysis and enable testing the relevance of GC–GC interactions by analyzing single GCs in a lymphoid organ. As variability among single GCs could arise due to other mechanisms, identifying other potential mechanisms that lead to such heterogeneity and investigating the contribution of each mechanism would help characterize the causes of GC–GC heterogeneity. Systematically injecting antibodies [29] at different concentrations and affinities for specific antigenic conditions would help differentiate the effects of antibody feedback from other mechanisms. Injecting antibodies or modulating the production of antibodies could alter the clearance of antigen from FDCs [54] but differential effects on immunodominant and subdominant epitopes would indicate the action of soluble antibodies by epitope masking rather than the antigen clearance.

In our previous study [47], we showed that antibody feedback is one of the mechanisms that is able to independently promote shutdown of GCs. However, we could not identify experiments that distinguish the action of soluble antibodies from the consumption of antigen by B cells. Experiments proposed to detect the existence of antigen limitation such as the measurement of pMHC density combined with observations such as variability identified among individual GCs in this study might help differentiate the action of soluble antibodies from antigen consumption by B cells.

In this study, we have assumed that antibodies produced are well-mixed throughout the organism. It remains to be explored in experiments whether antibodies secreted by plasma cells in the GC area result in a locally higher concentration of antibodies inside GCs. Direct diffusion of antibodies from a GC to neighboring GCs was also neglected in this study. While the mechanisms considered in the GC model are common to GCs in spleen and lymph nodes, initiation times and number of GCs in the asynchronous setting were estimated using experimental data from the spleen. We expect a similar qualitative effect of GC–GC interactions in lymph nodes, however whether the permeability and local concentration of antibodies differ in the GCs of spleen and lymph nodes needs further investigation. Relative contribution of different antibody isotypes towards antibody feedback due to differences in properties such as tissue permeability and half-lives is unclear and needs further investigation with a detailed model of isotype switching [55,56] that was not considered in the present investigation.

In the present study, we focused on the antibody mediated regulation of GCs by epitope masking. Other mechanisms of regulation of immune responses by antibodies have been identified such as Fcγ-mediated inhibition [57]. Moreover, under certain conditions, antibodies are capable of enhancing GC responses. For instance, depending on the subtype of IgG antibodies, GC responses is enhanced by Fcγ-ligation of dendritic cells [58]. There could also be additional levels of regulation in plasma cell secretion of soluble antibodies and stability of soluble antibodies that might modulate GC–GC interactions. Memory B cells have been shown to have upregulated expression of FcγRIIB and a decrease in the expression of FcγRIIB is associated with systemic lupus erythematosus [59]. Understanding the contribution of soluble antibodies via FcγRIIB—mediated action on B cells will facilitate development of treatment strategies for autoimmunity. Future research along these directions would be greatly beneficial to promote a better understanding of the natural regulation and therapeutic modulation of GCs by soluble antibodies.

## Figures and Tables

**Figure 1 cells-11-03680-f001:**
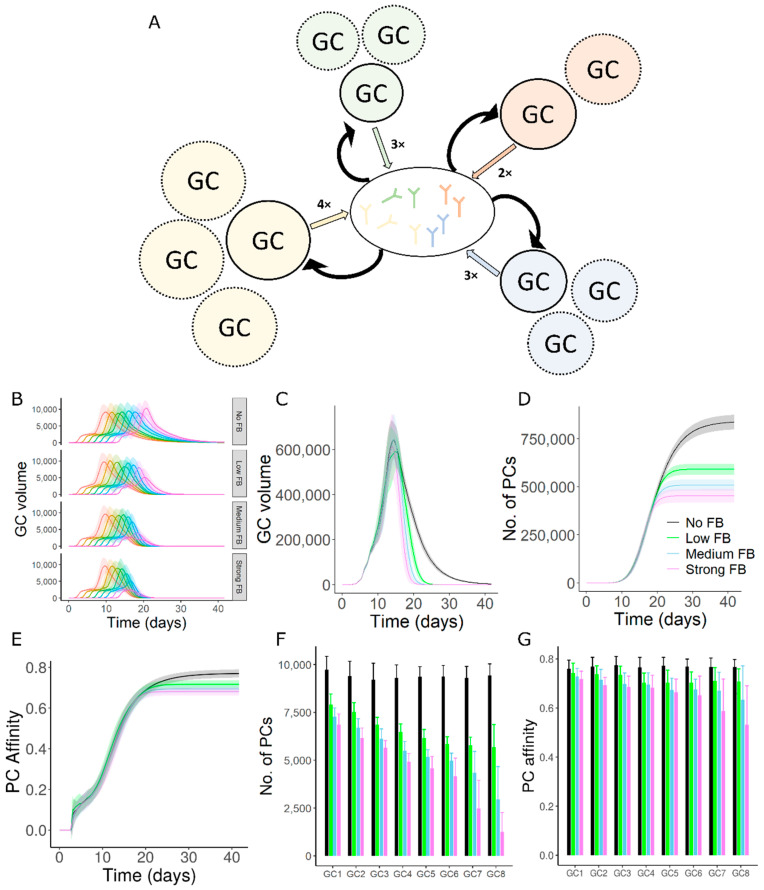
GC–GC interactions limit the magnitude of GC responses. (**A**) Schematic representation of GC–GC interactions in a multi-GC agent-based model. Colored arrows represent the production of soluble antibodies from individual GCs. Black arrows represent the feedback of soluble antibodies on simulated GCs. Solid circles represent simulated GCs and dotted circles show non-explicitly simulated GCs represented by every simulated GC. n in n× is the number of synchronous GCs to account for the antibody contribution of non-explicitly simulated GCs. Values of n shown in this figure are chosen for representation purpose only. See methods for values used in simulations. (**B**) Volume kinetics of simulated GCs. GCs were initialized asynchronously, and color code represents the timing of GC initiation. (**C**) Total volume of all GCs. (**D**) Plasma cell production from all GCs. (**E**) Affinity of plasma cells from all GCs. Readouts in C-E represent the sum of corresponding readouts of all simulated GCs weighted by the number of non-explicitly simulated GCs in each case. (**F**) Number of plasma cells produced from individual GCs. (**G**) Affinity of plasma cells from individual GCs. GCs 1–8 were sorted in the sequence of initiation. Inset in panel D shows the GC–GC interaction strength (see methods) and applies for panels C-G. A single epitope (shape space position 3333) was considered with founder cell positions chosen randomly anywhere in the shape space.

**Figure 2 cells-11-03680-f002:**
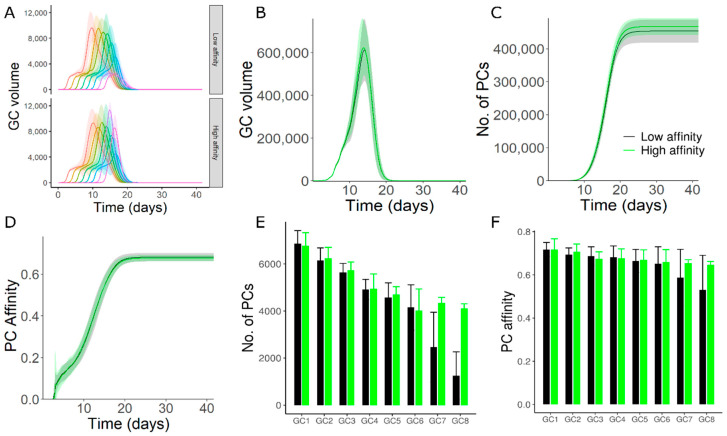
Simulation of late GCs with high affinity founder cells. (**A**) Volume kinetics of simulated GCs. GCs were initialized asynchronously and color code represents the timing of GC initiation. (**B**) Total volume of all GCs. (**C**) Plasma cell production from all GCs. (**D**) Affinity of plasma cells from all GCs. Readouts in B-D represent the sum of corresponding readouts of all simulated GCs weighted by the number of non-explicitly simulated GCs in each case. (**E**) Number of plasma cells produced from individual GCs. (**F**) Affinity of plasma cells from individual GCs. GCs 1–8 were sorted in the sequence of initiation. Inset in panel C applies for panels (**B**–**F**). In simulations labelled “Low affinity”, founder cells of GCs 1–8 were chosen randomly, and in “High affinity”, founder cells of GC7 and GC8 were chosen at a distance of 2 mutations from the optimal shape space position (3333). Strong GC–GC interaction strength (see methods) was considered in these simulations.

**Figure 3 cells-11-03680-f003:**
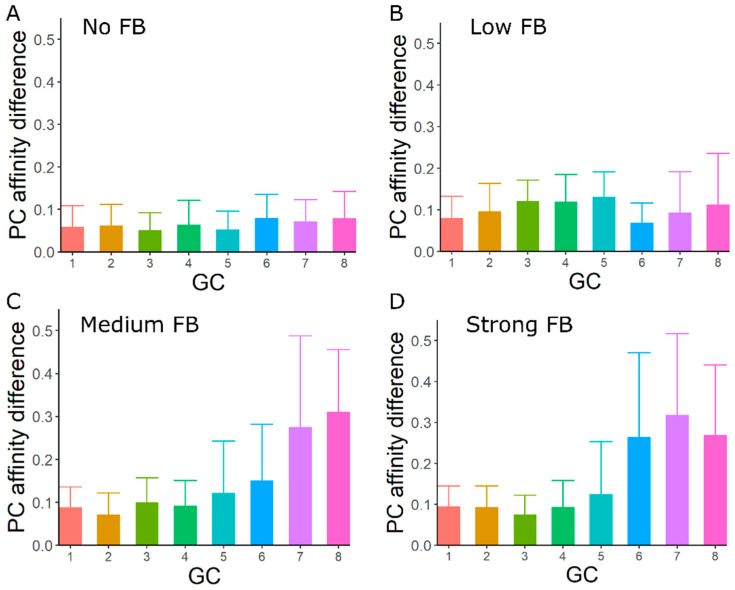
Differences in affinity maturation towards two different epitopes. (**A**–**D**) Differences in affinity of PCs between two epitopes for different strengths of GC–GC interactions (No FB, Low FB, Medium FB and Strong FB). The PC affinity difference was calculated as the absolute value of the difference between PC affinity to the two different epitopes. GCs were initialized asynchronously with two epitopes in equal proportion (shape space positions 3333 and 5555) and founder cell positions chosen randomly anywhere in the shape space. GCs 1–8 were sorted in the sequence of initiation.

**Figure 4 cells-11-03680-f004:**
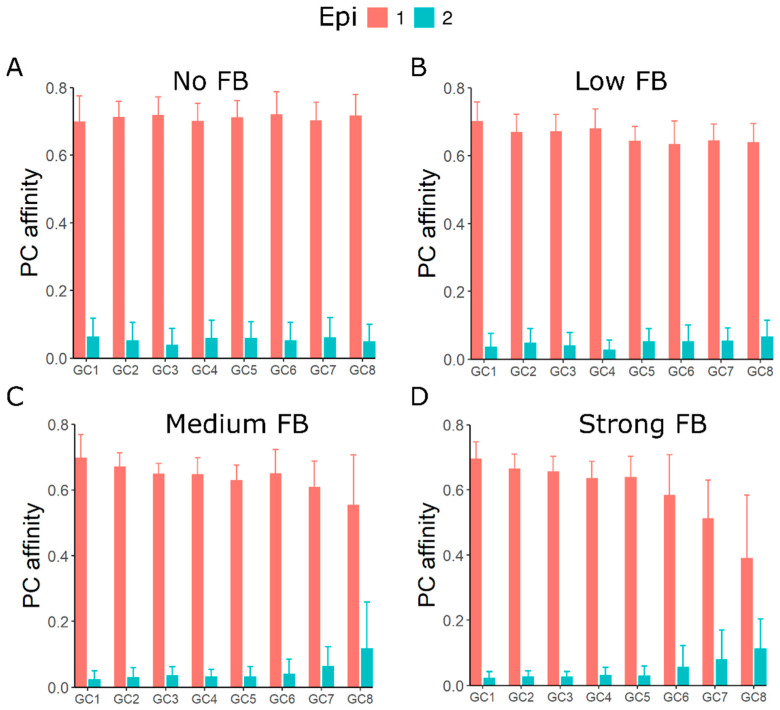
Affinity maturation to a rare and an immunodominant epitope. Affinity of plasma cells from asynchronously simulated GCs to two different epitopes, immunodominant (90%, Epi 1) and rare (10%, Epi 2), with different strengths of GC–GC interactions (**A**–**D**). Plasma cell affinity was calculated at the end of the GC simulation (40 days post immunization). Shape space positions of the two epitopes were 3333 (Epi 1) and 5555 (Epi 2) and founder cells were chosen randomly anywhere in the shape space. GCs 1–8 were sorted in the sequence of initiation. Colors represent the epitope to which affinity was calculated.

**Figure 5 cells-11-03680-f005:**
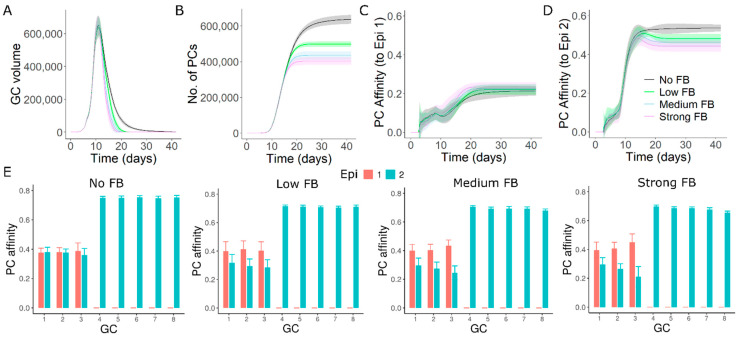
Interaction of GCs with different founder cell compositions. (**A**) Total volume of all GCs. (**B**) Plasma cell production from all GCs. (**C**) Affinity of plasma cells from all GCs to Epi 1. (**D**) Affinity of plasma cells from all GCs to Epi 2. Readouts in (**A**–**D**) represent the sum of corresponding readouts of all simulated GCs weighted by the number of non-explicitly simulated GCs in each case. (**E**) Affinity of plasma cells from individual GCs to different epitopes with different GC–GC interaction strengths (No FB, Low FB, Medium FB and Strong FB). GCs were simulated asynchronously initialized with two epitopes (shape space positions 2222 and 6666) in equal proportion. Founder cells of GCs 1–3 were chosen randomly anywhere in the shape space and founder cells of the rest of the GCs were chosen at a distance of 1 mutation from shape space position 6666. GCs 1–8 were sorted in the sequence of initiation.

**Figure 6 cells-11-03680-f006:**
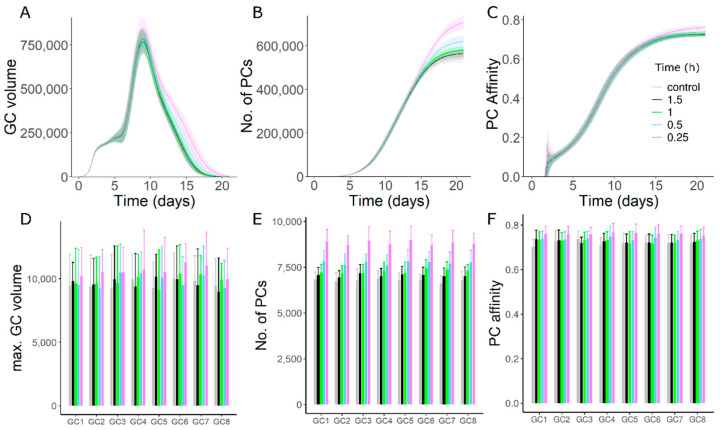
Persistent deposition of antigen in GC-FDCs. (**A**) Total volume of all GCs. (**B**) Total plasma cell production of all GCs. (**C**) Affinity of plasma cells from all GCs. (**D**–**F**) Maximum GC volume, number of plasma cells and affinity of plasma cells from individual GCs. Inset shows the time interval between consecutive antigen addition events to the FDCs during the GC reaction. No additional antigen was added after GC initiation in control simulations. GCs were synchronously initialized with a single epitope (shape space position 3333) and founder cell positions chosen randomly anywhere in the shape space. Low GC–GC interaction strength was considered in these simulations.

**Figure 7 cells-11-03680-f007:**
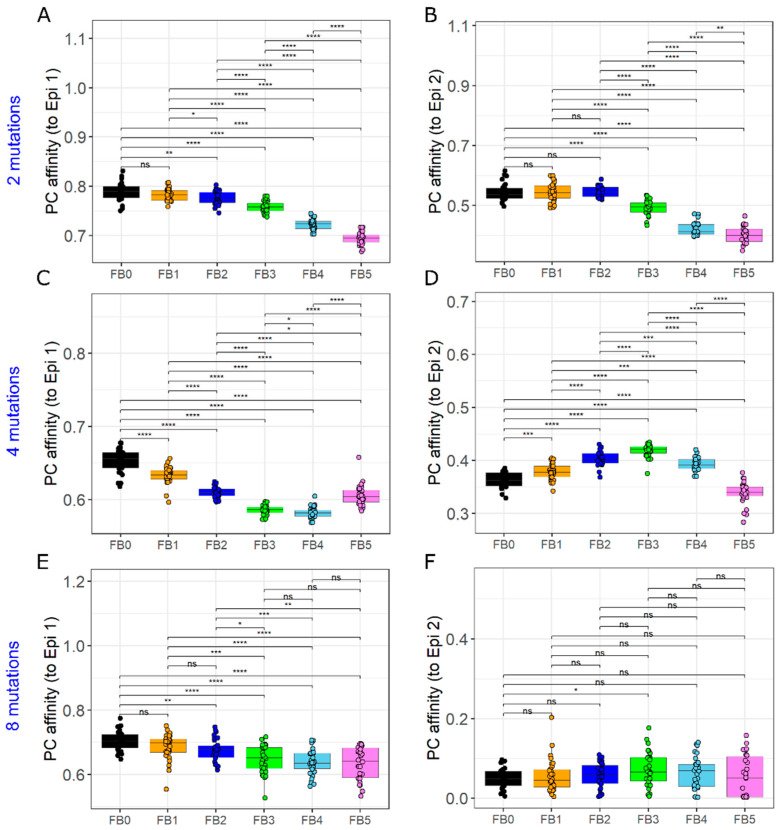
GC responses to antigenic variants. Affinity of plasma cells from all GCs to epitope 1 (**A**,**C**,**E**) and epitope 2 (**B**,**D**,**F**) at day 25 of the GC reaction. Mutation distance of epitope 2 with respect to epitope 1 is 2 (**A**,**B**), 4 (**C**,**D**) or 8 (**E**,**F**). Colors represent different GC–GC interaction strengths. FB0 represent simulations with no GC-GC interaction. FB1 and FB2 represent one-tenth and one-third of low interaction strength, respectively. FB3, FB4 and FB5 correspond to low, medium and strong interaction strengths, respectively. GCs were initialized with epitope 1 (shape space position 3333) and an antigenic variant (epitope 2) with mutation distance of 2, 4 or 8 (shape space positions 3344, 4444 and 5555, respectively) from epitope 1 was added from day 7 of GC reaction every 0.25 h. Founder cells of GCs were chosen randomly anywhere in the shape space. * = *p* < 0.05, ** = *p* < 0.01, *** = *p* < 0.001, **** = *p* < 0.0001, ns = not significant.

## Data Availability

C++ code and R scripts used in this study are available at https://doi.org/10.5281/zenodo.6568368 (uploaded on 21 May 2022).

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
