# Peer review of "Antibody Mediated Intercommunication of Germinal Centers"

_cells, 2022, doi:10.3390/cells11223680_

Round 1

Reviewer 1 Report

The study by Arulraj et al. used the previously developed model of a single germinal center (GC) to explore the role of antibody feedback from multiple GCs, with their synchronous or asynchronous onset, on the dynamics of individual GCs and overall germinal center reaction. The manuscript is well written, and the results are novel. Their studies have shown that GC-GC interactions through feedback from endogenous antibodies may affect the epitope recognition and dominance of antibody responses to different epitopes. They also explored the interplay of soluble antibodies and persistent deposition of antigen on follicular dendritic cells.

The comments below are mostly to clarify the model assumptions and application.

Lines 88-90: Could you please give the reasoning on why a shape space dimension was set to be equal to 4? The provided reference (Ref 35: Perelson & Wiegel, 1999) has a continuous space model that does not talk about four-dimensional space. Could you please explain where the number 4 was derived? What is the range used for each position in a four-digit code? How many mutations away from the optimal position would correspond to low, medium, and high affinity (in relation to lines 127-129)? Also, what are the random choices in four-dimensional space? It must be uniformly random, but how is the shape-space bounded? What happens at boundaries?

What was the mechanism used to end GC reaction when antibody feedback was turned off (“No FB” case in Figure 1B)? Line 145 says “the soluble antibodies were not allowed to mask the antigen on FDCs”. Do you still have antibody feedback from the individual GC?

Figure 1 and Figure S1 show that after an initial increase in GC volume there is a short plateau between days 2 and 6 (e.g. Figure S1) before a sharp increase. What are the potential reasons for this surprising behavior?

Section 3.4 and 3.5 use shape space positions 3333 and 5555 to define two different epitopes. Are these positions too close? Would we expect two randomly chosen epitopes to be more distant in the shape space? Will a B cell at position 4444 through some divisions/mutations have daughter cells specific to both epitopes? Could it affect the result shown in Figure S4B? Would you get the same or qualitatively different results using space positions further apart such as 3333 and 7777?

Minor points:

line 110: missing “(“

Eq. 4 does not show correctly on my computer, there seems to be a missing symbol

Legend to Figure 1: “insert in panel C” should be “insert in panel D”

Lines 250-251: the sentence is confusing, please clarify.

Line 296 please spell out “SRBC”

Line 248 suggestion to replace “Late formed GCs had higher affinity…” to “Late formed GCs tend to have a higher affinity…” as it is not always the case as shown in the right panel of Figure S4B

In several figures, error bars suggest potential negative numbers (e.g. in Figure 3 B, GCs #7-8 and Figure 4 C, GC #8). Please start the corresponding y-axes from zero.

Reviewer 2 Report

Comments

As a positive note, I really liked the figures, their quality is really good, well done.

Mathematical modeling-based simulations are based on assumptions and then we get conclusions from these assumptions. All conclusions authors found can be derived from assumptions with some thinking - e.g., all main statements in the abstract are obvious (see below). What is new brought by simulations? Authors need to step back and re-think what aspects of GC reaction they can falsify or discriminate between alternatives. For example, is there a level of Abs that early GC make that would not impact late GCs? Can this be measured experimentally, so the model can be tested/falsified?

The model of GC reaction is made in "vacuum", i.e., it is unclear if authors focus on LNs or spleen. I have concerns that plasma cell-produced Abs would be able to reach other GCs in the same LN given the lymph flow. If Abs are in circulation, they may not be able to reach the LNs either, especially if these are IgM. IgG is typically produced late and again, depending on the subclass, they may not penetrate that easily into the LN and specifically to GC. Assumption that Abs are "well-mixed" in the body seems unrealistic. In summary, while I understand that the experimental results of Zhang et al. JEM 2013 may be interesting, they are for spleen and with a high dose injection of IgG (and impact of injection was short-lived). Whether any of that can truly work with endogenously produced Abs remains a theoretical concept without any evidence.

The model is oversimplified in terms of Ab production by plasma cells - where do the cells make it? Probably in the blood. The model also ignores isotope switch.

How important are other model parameters on conclusions was not thoroughly investigated.

Which aspects of the GC reaction that model produces are relevant experimentally, i.e., can be compared to data? For example, strong FB reduces the PC number by about 40% (Fig 1D) and very minimally the PC affinity. Is that detectable by experiments given large variability in number of PCs made per immunization? If not, these results are pretty much irrelevant because cannot be tested. Same applies to many other "output" characteristics. Authors must carefully look and report features that can be compared with experiments. Otherwise, it is not science.

Minor comments

I would suggest to have abbreviation not in every caption but separately, next to keywords. Otherwise, these are just repeated wasting space.

Line 26 - Ab responses do not rely on generation of memory B cells. Perhaps you meant "result in"?

Commenting on the abstract:

"In this study, we performed in-silico simulations of interacting GCs and predicted that intense interactions by soluble antibodies limit the magnitude and lifetime of GC responses. "  -- this is obvious given the assumptions, no need for model.

"With asynchronous GC onset, we observed a higher inhibition of late formed GCs compared to early ones." - also obvious.

"We also predicted that GC-GC interactions can lead to a bias in the epitope recognition even 16 in the presence of equally dominant epitopes due to differences in founder cell composition or initiation timing of GCs." -- a bit more complex but can be guessed.

"These findings suggest that GC-GC interactions might be a contributing factor to the unexplained variability seen among individual GCs. " - so can million of other things. How can we discriminate between them? (i.e., falsify some)?

"We also show that the inhibitory effect of soluble antibodies could be counteracted by persistent deposition of antigen on FDCs, thus identifying a natural mechanism that could limit the antibody mediated feedback regulation." - again, if Abs mask Ag, adding more Ag eventually must help. Obvious.

[Same points can be applied in the text where similar "obvious" statements are made, including an obvious conclusion:

"Taken together, our results suggest that depending on the antigen complexity, antibody-mediated GC-GC interactions lead to differences in individual GC kinetics, quality and quantity of plasma cells in an initiation time dependent manner or a characteristic shift in affinity maturation towards specific epitopes."

Line 29-30 should include "LZ" to keep in structure with a sentence about "DZ".

Using V=10ml is not justified. In our experiments, we typically can get 0.5ml of blood from a B6 mouse.

eq(1) assumes that Abs are made in GC. I am not sure what is the relative contribution of plasma Abs from cells in the GC vs. circulation.

Assumed Ab production rates seem very small, <1 molecule per cell per hour. This paper (PMID:  25621280) discussed at length that current estimates of the Ab production rates by plasma cells are not reliable and vary orders of magnitude between different studies.

Results with memory B cells may need re-thinking. Experimental studies found that memory B cells express higher levels of FcGammaIIB receptor (e.g.,  16923849), so these cells will be more sensitive to IgG-mediated feedback.

The way deposition of additional Ag is modelled is unrealistic. It is impossible experimentally to deposit Ag on FDCs, one will administer the Ag as an injection. But then circulating Abs will neutralize most of it, so modeling must account for that.

Lines 424-432 - can you discriminate between these alternatives? That would be a good paper.
